# Kinematic Modeling and Experimental Study of a Rope-Driven Bionic Fish

**DOI:** 10.3390/biomimetics9060345

**Published:** 2024-06-07

**Authors:** Bo Zhang, Yongchen Huang, Zhuo Wang, Hongwen Ma

**Affiliations:** College of Mechanical and Electrical Engineering, Harbin Engineering University, Harbin 150001, China; zhangbo_heu@hrbeu.edu.cn (B.Z.); wangzhuo_heu@hrbeu.edu.cn (Z.W.); mahongwen@hrbeu.edu.cn (H.M.)

**Keywords:** bionic fish, wire drive, flexible spine, simulation

## Abstract

This paper presents a biomimetic fish robot featuring a flexible spine driven by cables, which integrates the cable-driven mechanism with a flexible spine. The drive system separates the body and tail fin drives for control, offering enhanced flexibility and ease in achieving phase difference control between the body and tail fin movements compared to the conventional servo motor cascaded structure. A prototype of the biomimetic fish robot was developed, accompanied by the establishment of a kinematic model. Based on this model, a control method for the biomimetic fish is proposed. Additionally, we introduce the concept of prestress to establish a numerical model for the biomimetic fish. Using multi-physical field simulation software, we simulate the two-dimensional autonomous swimming process of the biomimetic fish under different flapping frequencies and solve for its swimming characteristics as well as hydrodynamic properties. Both the simulation and experimental results validate the accuracy of our kinematic model.

## 1. Introduction

The ocean harbors abundant mineral resources, marine biological resources, and energy reserves, making the development of oceanic resources crucial for sustainable human progress. Underwater robotics research has garnered significant attention both domestically and internationally as a vital tool in this endeavor. Traditional underwater robots primarily employ motor drives due to their simple structure and convenient control; however, they suffer from drawbacks, such as low efficiency, limited mobility, and high noise levels. As human exploitation of the ocean accelerates alongside advancements in disciplines like biomimicry, fluid mechanics, and computer science, there is a growing need for underwater robots and vehicles that emulate fish by exhibiting high efficiency, low noise emissions, and flexible maneuverability [1,2,3].

The propulsion modes of fish can be categorized into two main types based on the differences in the source of fish movement and the manner in which fins are utilized: BCF (body and/or caudal fin) and MPF (median and/or paired fin) [4,5,6,7]. The body of BCF mode fish is soft and less rigid than that of MPF mode, and it utilizes the fluctuating curvature of the spine to drive the tail fin to oscillate and thus obtain a larger thrust, thus achieving a high swimming speed and propulsion efficiency. Currently, most of the BCF propulsion mode bionic fishes adopt jointed tandem mechanism [8,9,10,11].

There have been numerous studies conducted on biomimetic fish in the past, as shown in Appendix Table A1, resulting in significant advancements in the field of muscle–tendon-driven biomimetic fish. In 1994, David Barrett’s team successfully developed “Robo Tuna”, the first biomimetic robotic fish that accurately replicated the movement of Bluefin tuna by utilizing tail flapping for propulsion. This remarkable creation achieved a propulsion efficiency of 91% and reached a maximum swimming speed of 2 m/s (1.67 BL/s) [12,13]. In 2005, researchers at the University of Essex in the UK introduced “fish-G9”, a biomimetic carp with its tail driven by three servo motors arranged in series to realistically simulate fish-like movements [14,15]. Although it exhibited some flexibility, there remained notable gaps regarding efficiency, speed, and maneuverability. Consequently, they developed an improved robotic fish named “iSplash-II” [16,17], measuring 32 cm long and capable of generating high-speed propulsion through servo motor driving via high-frequency oscillation. From 2012 to 2017, Yong Zhong et al. employed a pulling line mechanism to achieve C-shaped and S-shaped oscillations of the tail fin, effectively simulating the flexibility observed in real fish bodies while maintaining exceptional maneuverability. Subsequent research focused on investigating how different driving curves influenced stability and cruising ability within biomimetic fish models [18,19,20]. In 2018, Yueqi Yang et al. investigated the fault-tolerant control problem of a multi-jointed machine fish with diverse tail fins. By employing fault-tolerant control methods to adjust the motion of the faulty machine fish, it was possible to enhance the stability and lifespan of the actual robot system even in damaged conditions [21]. In 2010, Robert J. Webster proposed that constant curvature kinematics could be decomposed into two independent submappings: one is universally applicable to all continuum robots, while the other is specific to this particular machine [22]. In 2018, Anand Kumar Mishra et al. introduced a kinematic model and cantilever beam model to separately describe the positions and stiffness of basic modules in a modular continuous arm [23]. This approach can be utilized for a kinematic analysis of biomimetic fish in this study. In 2023, Changlin Qiu et al. presented a unique tendon-driven structure incorporating torsional springs, enabling the realization of a passive tail fin with variable stiffness through motors and springs [24].

The wire-driven flexible spinal column of the biomimetic fish mechanical structure is characterized by its simplicity, facilitating enhanced control over system stability during the design phase. It exhibits a high degree of under-actuation, rendering it suitable for executing large-scale flexible movements. By employing a single servo motor to drive a group of wires, the number of servomotors required for controlling the biomimetic fish is reduced, thereby alleviating control complexities. Consequently, this biomimetic fish holds significant practical value and can be effectively utilized in diverse applications such as water quality monitoring, underwater exploration, and military reconnaissance in complex terrains like trenches and rocky reefs.

The paper initially presents the comprehensive design of the wire-driven flexible vertebral artificial fish, followed by the establishment of motion models for both its body and tail fin. Based on these models, a control method is devised. Subsequently, a comparison is made between the wire-driven flexible vertebral mechanism and the joint serial mechanism in terms of their respective fish body wave curves. Through this comparison, the superiority of this type of mechanism over the joint serial type mechanism is demonstrated.

## 2. Design Method of Bionic Fish

The primary propulsive force in the swimming process of fishes with a crescent tail fin propulsion mode is primarily attributed to the fluctuation of the spinal curve. The presence of a flexible fish body and swinging caudal fin indicates the occurrence of traveling waves, known as fish body waves, which propagate from the back neck to the caudal stalk during swimming [25,26]. These fish body waves originate at the center of inertia force located at the back neck and extend towards the end of the tail handle. The functions performed by fish body waves are as follows:(1)$$y_{body}(x,t)=(c_{1}x+c_{2}x^{2})\sin(kx+wt),$$

The body oscillation of the bionic fish is determined by the coefficients in the body wave equation. The maximum designed amplitude of body oscillation for this study’s bionic fish is 0.1 times its body length.

The complete system consists of the head motion control component, body motion executive component, and the fin component. Within the fish head, the battery, drive mechanism, and controller are housed. The bionic fish’s flexible spine is constructed using elastic material. Along this flexible spine lies a multi-segmented shell with an outer surface covered in skin and an inner shell perforated to accommodate the driving line. The pectoral fin enables the bionic fish to float or sink as required for its functionality. Figure 1 shows the overall structure composition of the bionic fish.

The movement composition: the fish head movement control component regulates the motion execution component of the fish body through precise movement control. The internal servo motor of the fish head is divided into two groups: the fish body and tail fin rudder group, as well as the pectoral fin rudder group. A winding mechanism connects the fish body and tail fin rudder group to the motion execution part, enabling control over both individual movements of the fish body and composite movements of its tail fin. Specifically, two groups of servo motors govern the fish body, while one group controls the fish tail. Each set of servo motors is intricately wound with a corresponding set of wires via a winding mechanism, allowing for wire tension adjustments by rotating each servo motor to precisely regulate both fish body and tail fin movements. Additionally, a coupling mechanism links to pectoral fin shafts facilitating rotation control over pectoral fins.

The shell size of biomimetic fish was determined by referencing the shape ratio of gourd fish [27]. The key parameters for the morphology and dimensions of biomimetic fish are presented in Table 1.

The anterior portion of the fish head shell contour exhibits an elliptical shape, while the posterior portion follows a parabolic curve. The mathematical equation representing this curvature is as follows:(2)$$\left\{\begin{array}{l} y=\pm \frac{D}{2L_{c}}\sqrt{{L_{c}}^{2}-x^{2}}\;\;\;\;\;\;0\leq x\leq 280\\ y=\pm \frac{D}{2}(1-\frac{x^{2}}{{L_{r}}^{2}})\;\;\;\;\;\;280\leq x\leq 600 \end{array}\right.,$$ where D is the diameter of the largest section of the revolving body (elliptical short axis); $L_{c}$ is length of the first half of the fish head; $L_{r}$ is the length of the posterior half of the body. In Equation (2), D is 54 mm, $L_{c}$ is 280 mm, $L_{r}$ is 320 mm. The shell is made of a photosensitive resin material by 3-D printing.

The joints of the fish body can be categorized into active and follower joints based on their distinct functions. Active joints are directly controlled by the pull line, while follower joints are evenly distributed along the flexible spine. These joints possess an elliptical cross-sectional area, exhibiting a decreasing distribution from head to tail. Each joint has a uniform thickness of 10 mm, resulting in an overall outline of the fish body resembling a spindle formed by parabolic curves. In theory, as the number of follower joints approaches infinity, the shape of the driving line will converge towards a circular arc representing the flexibility of the spine. The static center of gravity for this biomimetic fish was calculated using three-dimensional design software and adjusted using a gravitational square method to align with its floating center.

The control system consists of an upper computer, a USB serial port module, an MCU, a power supply module, and an executive rudder. The upper computer transmits control parameters to the lower computer through the USB serial port module and subsequently sends driving signals to the rudder via the lower computer in order to achieve the fish tail swing, as illustrated in Figure 2.

A HV-CLS-1323-Blade steering gear is used to drive the cable, a lithium battery is used as power supply, and ARM is used as microcontroller.

## 3. Kinematic Analysis of Bionic Fish

### 3.1. Kinematics Model and Control Method

At present, there are three common bionic fish motion control methods: a motion model fitting method based on the fish body wave curve, a control method based on an equivalent simplified hydrodynamic model and a model method based on CPG [28,29]. In this paper, a motion model fitting method based on the fish body wave curve is adopted for a flexible fish body driven by a pull-line and a rigid tail fin controlled by separation.

The kinematics model of the fish body and the motion analysis of fish body oscillation driven by a pull-wire can refer to the kinematics analysis of the flexible manipulator with pull-wire. The kinematics model of the fish body is established based on a piecewise constant curvature model. The piecewise constant curvature model has been successfully applied to the kinematics analysis of a series of flexible continuum robots, such as mobile phone robots and bionic elephant nose manipulators [30].

The motion of the flexible body of the robotic fish was analyzed using the segmented constant curvature method [31]. During motion, the flexible spine along the axis was divided into *n* equal-length small arcs. In each small arc, intermediate variables were introduced to express the bending and twisting states of the driving line under different stretching conditions. These intermediate variables include $r_{i}$, representing the center arc curvature radius; $\theta_{i}$, representing the center arc curvature angle; $\Phi_{i}$, representing the bending plane angle; and $\Phi_{i}=0^{{}^{\circ}}$ if only considering two-dimensional lateral movement relative to the head–tail axis of the fish body. In Figure 3, the upper and lower cross-sections of each small arc i(0≤i≤n) are shown with the established inertial coordinate systems $o_{i}-x_{i}y_{i}z_{i}$ and $o_{i+1}-x_{i+1}y_{i+1}z_{i+1}$. The geometric model is depicted in Figure 4 as well. When in its initial undeformed state, with a flexible body deformation occurring afterwards, $z_{i}$ axis aligns with the center arc of fish body while $y_{i}$ axis remains parallel to its left–right axis.

The initial length of the four driving ropes is denoted as L, with their respective variations in length represented by $q_{1},\;q_{2},\;q_{3},\;q_{4}$. At any given time, the lengths of the four driving ropes are indicated as $l_{1},\;l_{2},\;l_{3},\;l_{4}$:(3)$$\left\{\begin{array}{l} l_{1}=L-q_{1}\\ l_{2}=L-q_{2}\\ l_{3}=L-q_{3}\\ l_{4}=L-q_{4} \end{array}\right.,$$

The length l of the central arc can be determined based on the lengths of four driving lines at any given time.
(4)$$l=\frac{l_{1}+l_{2}+l_{3}+l_{4}}{4},$$

The cross-section (longitudinal section) perpendicular to the $z_{i}$ axis of the flexible fish body is elliptical. However, for the purpose of motion analysis, we focus on studying the cross-section of the circumference formed by the four driving lines within this elliptical cross-section. All four driving lines intersect with this circumference. Consequently, we can consider the fish body as a circular platform with a gradually decreasing radius in its longitudinal section. Assuming a constant sectional curvature, Figure 4 illustrates both the distribution of driving lines and the bending plane angle $\Phi_{i}$ in section i.

The endpoints of the four driving lines, located on the circumference of section i in the equivalent flexible fish body, form rectangular shapes. The angle between the plane of the central arc (indicated by a red mark) and the $y_{i}$ axis is denoted as $\Phi_{i}$. $R_{i}$ represents the radius of the cross-section circle. It is known that $R_{\max}$ denotes the initial cross-section radius of the equivalent flexible fish body, while $R_{\min}$ represents its end cross-section radius. Based on the Figure 4 geometric relationship, we can express section i radius $R_{i}$ as follows:(5)$$R_{i}=R_{\max}+\frac{i}{n}(R_{\min}-R_{\max}),$$

The curvature of the four driving lines in each segment is identical to that of the central arc, as depicted by the geometric relationship illustrated in Figure 4. Consequently, it becomes feasible to determine the curvature radius $r_{ij}(j=1,2,3,4)$ for each driving line.
(6)$$\left\{\begin{array}{l} r_{i1}=r_{i}-R_{i}\cos(\frac{\pi }{4}+\phi_{i})\\ r_{i2}=r_{i}-R_{i}\cos(\frac{\pi }{4}-\phi_{i})\\ r_{i3}=r_{i}+R_{i}\cos(\frac{\pi }{4}+\phi_{i})\\ r_{i4}=r_{i}+R_{i}\cos(\frac{\pi }{4}-\phi_{i}) \end{array}\right.,$$

Given the known radius of curvature, it is possible to mathematically express the arc length of each driving line within every segment.
(7)$$\left\{\begin{array}{l} \frac{l_{1}}{n}=\theta_{i}[(r_{i}-R_{i}\cos(\frac{\pi }{4}+\beta )]\\ \frac{l_{2}}{n}=\theta_{i}[(r_{i}-R_{i}\cos(\frac{\pi }{4}-\beta )]\\ \frac{l_{3}}{n}=\theta_{i}[(r_{i}+R_{i}\cos(\frac{\pi }{4}+\beta )]\\ \frac{l_{4}}{n}=\theta_{i}[(r_{i}+R_{i}\cos(\frac{\pi }{4}-\beta )] \end{array}\right.,$$

The equation can be solved to obtain the solution:(8)$$\left\{\begin{array}{l} \theta_{i}=\frac{\sqrt{{\left(l_{1}-l_{2}\right)}^{2}+{\left(l_{2}-l_{3}\right)}^{2}+{\left(l_{3}-l_{4}\right)}^{2}+{\left(l_{4}-l_{1}\right)}^{2}}}{2nR_{i}}\\ r_{i}=\frac{(l_{1}+l_{2}+l_{3}+l_{4})R_{i}}{2\sqrt{{\left(l_{1}-l_{2}\right)}^{2}+{\left(l_{2}-l_{3}\right)}^{2}+{\left(l_{3}-l_{4}\right)}^{2}+{\left(l_{4}-l_{1}\right)}^{2}}}\\ \phi_{i}=\arctan(\frac{l_{4}-l_{2}}{l_{3}-l_{1}})-\frac{\pi }{4} \end{array}\right.,$$

Considering the mapping relationship between the input variable q and intermediate variables $\Phi_{i},\;r_{i},\;\theta_{i}$ for a flexible fish body, the transformation of circular arc sections in the i segment along the body coordinate system can be interpreted as an ordered sequence of translations and rotations. By incorporating the D–H method, we can derive the second transformation matrix from the initial coordinate system $o_{0}-x_{0}y_{0}z_{0}$ to the i coordinate system $o_{i}-x_{i}y_{i}z_{i}$. The transformation from $o_{i}-x_{i}y_{i}z_{i}$ to $o_{i+1}-x_{i+1}y_{i+1}z_{i+1}$ can be divided into five steps when there is no bending plane angle $\Phi_{i}=0{}^{\circ}$. In the case of the absence of torsion in the flexible fish body, Figure 5 illustrates this transformation process.

The initial position of the coordinate system $o_{i}-x_{i}y_{i}z_{i}$ is denoted by $o_{0}-x_{0}y_{0}z_{0}$, while the final position of the coordinate system $o_{i+1}-x_{i+1}y_{i+1}z_{i+1}$ after motion is represented by $o_{5}-x_{5}y_{5}z_{5}$. The transformation from the coordinate system $o_{i}-x_{i}y_{i}z_{i}$ to $o_{i+1}-x_{i+1}y_{i+1}z_{i+1}$ can be achieved through the following steps:

Rotate the angle $\Phi_{i}$ around the $z_{0}$ axis and −π/2 around the $x_{0}$ axis in a clockwise direction (positive). At this moment, it aligns with the $z_{0}$ and $y_{0}$ axes, as well as the $x_{1}$ and $x_{0}$ axes.Perform a rotation of $\theta_{i}/2$ around the $z_{1}$ axis, followed by a π/2 rotation around the $x_{1}$ axis.Translate along the $z_{2}$ axis by length $d_{3}$ while rotating −π/2 around the $z_{2}$ axis to align with the $x_{3}$ axis.Rotate by an angle of $\theta_{i}/2$ around the $z_{3}$ axis and π/2 around the $x_{3}$ axis. This results in alignment between the $x_{4}$ and $x_{5}$ axes, as well as between the $z_{4}$ and $z_{5}$ axes.Finally, rotate by an angle $-\Phi_{i}$ around the $z_{5}$ axis.

In the process of D–H transformation, the rotation angle around the z axis is denoted as β, the translation distance along the z axis is represented by d, the translation distance along the x axis is indicated as a, and finally, the rotation angle around the x axis is denoted as α. The parameters for D–H transformation are presented in Table 2.

The homogeneous transformation matrix from the coordinate system $o_{i}-x_{i}y_{i}z_{i}$ to the coordinate system $o_{i+1}-x_{i+1}y_{i+1}z_{i+1}$ is represented by ${\;}^{i}T_{i+1}$, which signifies a sequence of successive motions. All transformations are based on the current coordinate system and involve consecutive right-multiplication matrices. The operators c(⋅) and s(⋅) in the matrix denote cos(⋅) and sin(⋅) functions, respectively. The findings are as follows:(9)$$\begin{array}{l} {\;}^{i}T_{i+1}=Rot(Z,\beta )\times Rot(X,\alpha_{1})\times Rot\left(Z,\theta_{1}\right)\times Rot\left(X,\alpha_{2}\right)\times \\ Tran\left(Z,d_{3}\right)\times Rot\left(X,\alpha_{3}\right)\times Rot\left(Z,\theta_{4}\right)\times Rot\left(X,\alpha_{4}\right)\times Rot(Z,-\beta )\\ =\left[\begin{array}{cccc} c^{2}\beta_{i}(c\theta_{i}-1)+1 & s\beta_{i}c\beta_{i}(c\theta_{i}-1) & c\beta_{i}s\theta_{i} & r_{i}c\beta_{i}(1-c\theta_{i})\\ s\beta_{i}c\beta_{i}(c\theta_{i}-1) & s^{2}\beta_{i}(c\theta_{i}-1)+1 & s\beta_{i}s\theta_{i} & r_{i}s\beta_{i}(1-c\theta_{i})\\ -c\beta_{i}s\theta_{i} & -s\beta_{i}s\theta_{i} & c\theta_{i} & r_{i}s\theta_{i}\\ 0 & 0 & 0 & 1 \end{array}\right] \end{array},$$

The coordinate transformation employed in this study satisfies the chain rule, enabling us to obtain the positive kinematics homogeneous transformation matrix T from the initial point to the end point on the centerline of the flexible fish body. This is achieved by multiplying each small section’s homogeneous transformation matrix ${\;}^{i}T_{i+1}$, which represents both upper and lower sections of a circular arc, starting from the top coordinate system of the first section and ending at the bottom coordinate system of the last section.

By utilizing the mapping correlation between the length of the driving line and the intermediate variable, the positional and attitudinal data of any endpoint within the n segment arc of the fish body’s central axis can be obtained by incorporating q into T. The spatial location and orientation of points along the fish body’s central axis can be represented using a three-dimensional coordinate point $p(p_{x},p_{y},p_{z})$ and three Euler angles (α,β,γ).
(10)T=T10T21…Ti+1i…Tnn−1=nxoxaxdxnyoyaydynzozazdz0001,

The spatial coordinates $p(p_{x},p_{y},p_{z})$ of the points along the central axis of the fish body extracted from T are as follows:(11)$$\left\{\begin{array}{l} p_{x}=d_{x}\\ p_{y}=d_{y}\\ p_{z}=d_{z} \end{array}\right.,$$

The expression of the spatial Euler angles (α,β,γ) at the centroid of the fish body is as follows:(12)$$\left\{\begin{array}{l} \alpha =\arctan\frac{a_{y}}{a_{z}}\\ \beta =\arctan\frac{a_{x}\cos\alpha +a_{y}\sin\alpha }{a_{z}}\\ \gamma =\arctan\frac{-n_{x}\sin\alpha +n_{y}\cos\alpha }{-o_{x}\sin\alpha +o_{y}\cos\alpha } \end{array}\right.,$$

The correlation between the position and velocity of the driving line and the points on the fish body’s centerline is established. By regulating the amount and the speed of the pulling line, it is possible to manipulate both the swing amplitude and the frequency of the fish body, thereby enabling a thorough investigation into its propulsion performance. Under steady-state conditions, $q=|q_{1}|=|q_{2}|=|-q_{3}|=|-q_{4}|$. The relationship between q and $d_{y}$ at the end of fish body can be obtained through MATLAB R2018a simulation as depicted in Figure 6.

From the image, it is evident that the variable q undergoes stretching or shrinking from 0 mm to 5 mm. The oscillation amplitude $d_{y}$ exhibits an increase from 0 mm to 53.5492 mm. A linear relationship between q and $d_{y}$, approximately given by $d_{y}=kq$ with k=10.75 obtained through MATLAB’s linear fitting analysis, can be observed. Assuming R as the winding radius of the rudder winding wheel and θ as the angle of rudder turning from zero, we can establish a mapping relationship between the rudder angle and the swing amplitude at the fish body’s end since q=Rθ holds true. Analogously, it can be inferred that a linear correlation exists between the rudder angle and the swing amplitude at all points along the fish body.

The motion model of the tail fin is regulated by an intermediate drive line assembly. The tail fin is securely fastened in the slot of the tail handle joint using a bolt and nut mechanism. The tail handle joint is interconnected with the fish body joint through a hinge mechanism. The mechanical configuration of the tail fin can be observed from a top view, depicting three distinct motion states as illustrated in Figure 7.

In the initial state, the lengths of the tail fin driving lines are denoted as $l_{5}$ and $l_{6}$, while the lengths of the tail fin driving lines extending beyond the body are represented by ${l_{5}}^{\prime }$ and ${l_{6}}^{\prime }$. The variations in length are indicated as $q_{5}$ and $q_{6}$. When only the tail fin swings, the lengths of its driving lines exceeding the body part become ${l_{5}}^{\prime \prime }$ and ${l_{6}}^{\prime \prime }$. In this initial state, a is defined as 15.13 mm, representing the distance from center O to the end hole of the caudal handle joint; b is defined as 21.4 mm, indicating center O distance to a hole on one side of the fish body’s end joint; β remains constant at an angle value of 45°; θ represents rotation angle of caudal fin around hinge. Figure 8 illustrates the geometric relationship depicting length variation in the oscillating drive line for the tail fin.

The winding wheel consists of $l_{5}$ and $l_{6}$, where the absolute values of $q_{5}$ and $q_{6}$ are identical but with opposite signs. Figure 8 illustrates the geometric relationship between the rotation of the tail shank around the hinge and the driving line. By applying cosine to ${l_{5}}^{\prime \prime }$ and ${l_{6}}^{\prime \prime }$, the following can be determined:(13)$$\left\{\begin{array}{l} {l_{5}}^{\prime }=\sqrt{a^{2}+b^{2}-2ab\cos\beta }\\ {l_{5}}^{\prime \prime }=\sqrt{a^{2}+b^{2}-2\cos(\beta +\theta )ab} \end{array}\right.,$$
(14)$$q_{5}={l_{5}}^{\prime \prime }-{l_{5}}^{\prime },$$
(15)$$q_{5}=-q_{6},$$

During the actual motion of bionic fish, the tail fin rotates around the hinge axis and synchronously oscillates with the fish body. Therefore, $q_{5}$ and $q_{6}$ should encompass the variation in the tail fin in response to fish body translation. The driving lines for tail fins, $l_{5}$ and $l_{6}$, are positioned within the central region of the group of fish body driving lines. The variation of $l_{5}$ with respect to fish body translation, is represented by $(q_{1}+q_{2})/2$, while that of L6 during the fish body movement is denoted as $(q_{3}+q_{4})/2$. The variations in these lines can be described as follows:(16)$$\left\{\begin{array}{l} q_{5}=\frac{q_{1}+q_{2}}{2}+({l_{5}}^{\prime \prime }-{l_{5}}^{\prime })\\ q_{6}=\frac{q_{3}+q_{4}}{2}+({l_{6}}^{\prime \prime }-{l_{6}}^{\prime }) \end{array}\right.,$$

Let $\frac{q_{1}+q_{2}}{2}=0$, and the inverse function can be solved by MATLAB simulation:(17)$$\theta =\arccos(\frac{68597}{64680}-\frac{5\cdot (q_{5}+15.13{)}^{2}}{3234})-\frac{\pi }{4},$$

The relationship between $0<Q_{5}<b$, $Q_{5}$ and the angle θ of the tail fin around the hinge is well-established, as depicted in Figure 9.

From the observation of Figure 9, it is evident that the pendulum angle exhibits a nearly linear relationship with the driving line variable within the range of 0° to 40°, characterized by a slope of k=4. The bionic fish tail fin achieves a maximum pendulum angle of 30°. Consequently, precise control over the tail fin angle can be achieved by manipulating the steering gear angle in accordance with this established correlation.

According to this relationship, the program for the motion modes of bionic fish is designed. By adjusting the slave action time, the swing frequency of bionic fish can be modified in the case of a 100 mm body swing and a 1 Hz swing frequency.

### 3.2. Kinematic Simulation Results and Discussion

The fish body wave equation is:(18)$$x_{body}(z,t)=(c_{1}z+c_{2}z^{2})\sin(kz+wt),$$

According to the kinematics model proposed in this study for the flexible spine bionic fish body centerline, the fish body centerline is fitted with a wave equation using MATLAB. The obtained solution parameters are $c_{1}=0.005,\;c_{2}=0.00056,\;k=2\pi /6000$. In the fish body coordinate system, at time instances t=T/8, t=T/4, t=3T/8, t=5T/8, t=3T/4, t=7T/8; a comparison is made between the cruise mode’s fish body centerline and the corresponding fish body wave curve, depicted in Figure 10. The color line represents the body wave curve while the black dotted line depicts the trajectory of the bionic fish’s centerline.

The fitting calculation reveals that the parameters of body wave indicate a wavelength (λ) of 6000 mm, with a relative wavelength (R) of 1/20 for the bionic fish body oscillation proposed in this study. Notably, this R value significantly deviates from the gourd family’s relative wavelength (R=1/3). This discrepancy can be attributed to the C swing mode adopted by the bionic fish, which is contingent upon its body length.

The higher the fitting degree between the centerline locus of the bionic fish body and the body wave, the higher the flexibility and motion flexibility of the bionic fish body [32]. The centerline of the joint series biomimetic fish (including steering gear series, connecting rod series–parallel, pull-wire drive joint series) can be equivalent to the swing chain mechanism with n rigid connecting rods connected by hinges. Let the lengths of each rod be $l_{1},l_{2}\ldots$*l_n_*, in the plane composed of the left and right axes and the head and tail axes, and its corresponding endpoint coordinates are $\left(x_{0},y_{0}),(x_{1},y_{1})\;\ldots \;(x_{n},y_{n}\right)$.

The bionic fish in this project is driven by three actuators, therefore we have chosen to compare it with a three-joint bionic fish driven by the same actuator. In this section, we have selected T/8, 3T/16, T/4 as parameters to calculate the fitting degree of the bionic fish and the fish body wave curves of both the flexible spine mechanism driven by a cable and the joint-type tandem mechanism, respectively. The results of these fittings are presented in Figure 11, Figure 12 and Figure 13. The area enclosed by the fish body centerline and the envelope area of the fish body wave curve for both mechanisms are denoted as $S_{1}$ and $S_{2}$.

The integration of the centerline of the flexible spine fish body and the envelope area of the fish body wave curve is presented in Table 3.

The results presented in Table 3 demonstrate a clear inverse relationship between the envelope area of the fish body wave and the swing amplitude of the fish body, with an increase in swing amplitude leading to a decrease in envelope area. Conversely, an increase in swing amplitude is observed to result in an increase in the envelope area of the fish body wave. Notably, for small swings, the joint-type mechanism exhibits superior fitting with the fish body wave compared to the pull-line-driven flexible spine mechanism. However, as the fish body swing increases, it becomes evident that the pull-line-driven flexible spine mechanism achieves better alignment with the fish body wave than its joint-type counterpart. Consequently, we can conclude that cable-driven flexible spine mechanisms are more suitable for accurately simulating large swinging motions exhibited by fishes.

## 4. Analysis of Bionic Fish Dynamics

### 4.1. Numerical Analysis of Autonomous Swimming of Bionic Fish

The research in Section 3.1 yields the conclusion that the movement equation of the biomimetic fish’s centerline can be derived, and the fish body contour parameters can be parameterized by measuring the coordinates of characteristic points on its contour. Consequently, a mathematical model for the two-dimensional autonomous swimming of the biomimetic fish is established. Based on this model, an analysis is conducted to investigate the dynamics of autonomous swimming at different flapping frequencies, providing a foundation for optimizing motion control strategies. In this chapter, numerical simulation methods are employed and the concept of pre-stress $E^{0}$ is introduced to simulate bending deformation resulting from left and right flapping motions of the fish body. The COMSOL5.4 software’s FSI interface reproduces autonomous swimming behavior in a stationary fluid field.

The contour of the bionic fish in the x–y plane is illustrated in Figure 14a, and a total of 30 feature points are uniformly selected along its contour. By utilizing the coordinates of these feature points, MATLAB is employed to parameterize and represent the external contour of the bionic fish (including the tail fin), as depicted in Figure 14b.

The contour equation of the fish body shape is denoted as c(x), wherein the geometric modeling process of the COMSOL software incorporates a blunt circular arc representing the fish head, characterized by a radius of r = 30 mm.
(19)$$y=c(x)=a_{1}x^{5}+a_{2}x^{4}+a_{3}x^{3}+a_{4}x^{2}+a_{5}x,$$

The data were fitted using MATLAB, resulting in a fifth-degree polynomial c(x) with the following coefficients: $a_{1}=-1.628\times 10^{-11},\;a_{2}=2.025\times 10^{-8},\;a_{3}=-8.739\times 10^{-6},\;$ $a_{4}=1.063\times 10^{-3},\;a_{5}=0.244$.

The equation representing the centerline of the fish body is denoted as h(x,t), where H represents the maximum thickness of the fish body projected onto the x−y plane. The values of $c_{1},\;c_{2},\;k,\;w$ can be found in Section 3.2, while $t_{0}$ signifies the initial movement time. In addition to incorporating the fish body wave curve equation from Section 3.2, we introduce a new term $1-e^{\left(t/t_{0}\right)}$ to appropriately adjust for the initial timing of movement.
(20)$$h(x,t)=(c_{1}x+c_{2}x^{2})\sin(kx+wt)(1-e^{\left(t/t_{0}\right)}),$$

The function h(x,t) represents the lateral displacement (along the y axis) of the fish body’s centerline, while the pre-strain $E^{0}$ emphasizes the generation of fish movements rather than forces. The introduction of pre-strain $E^{0}$ allows for simulating the autonomous swimming of biomimetic fish in a flow field, experiencing counteracting forces. The relationship between the curvature of the fish body’s centerline, h(x,t), and pre-strain $E^{0}$ can be expressed as follows:(21)$$E_{xx}^{0}=-y\frac{\partial^{2}h(x,t)}{\partial x^{2}},$$

The hydrodynamic fish–fluid (water) coupled system can be divided into three distinct physical domains: $\Omega_{i}$ represents the boundary where fluid and solid are coupled, $\Omega_{s}$ represents the solid domain, and $\Omega_{f}$ represents the fluid domain.

The simulation of the fluid–structure coupling process in a two-dimensional plane flow field, which mimics the movement of a biomimetic fish, involves a two-way boundary coupling problem between an active flexible deformable body and the fluid [33]. The deformation of the solid body induces fluid flow, while the flow of the fluid simultaneously causes the deformation of the solid body. A distinguishing feature of this fluid–structure coupling problem is that it entails the significant deformation and displacement of the solid boundary. In addressing large deformation problems, either the Lagrangian description method (commonly used in solid mechanics) or the Eulerian description method (mainly used in fluid mechanics) is typically employed. In 1964, Noh proposed an arbitrary Lagrangian–Eulerian (ALE) method [34] to solve a finite difference-based problem involving a two-dimensional moving boundary for fluids. Unlike global solution methods, ALE description necessitates solving nested time domains for both the solid domain $\Omega_{x}$ and the fluid domain $\Omega_{x}$; hence, distinct governing equations are utilized to describe both solids and fluids [35].

The mass conservation equation, known as the continuity equation, states that the rate of mass outflow from a control volume is equal to the rate of mass inflow into the control volume. In a spatial coordinate system, this equation can be expressed differentially as follows:(22)$${\dot{\rho }}_{f}+\rho_{f}div(v_{f})=0\;\;\;\;\;\mathrm{in}\;\Omega_{x},$$

In this equation, $\rho_{f}$ denotes the fluid density, while $v_{f}$ represents the fluid velocity within the spatial coordinate system.

The momentum equation can be expressed in tensor notation as follows:(23)$$\rho_{f}{\dot{v}}_{f}+\rho_{f}(v_{f}\cdot \nabla )v_{f}=div\Gamma +F\;\;\;\;\;\mathrm{in}\;\Omega_{x},$$

The left-hand side of the equation represents the rate of momentum change per unit volume of the fluid, where $\rho_{f}{\dot{v}}_{f}$ denotes the local acceleration term. $\rho_{f}(v_{f}\cdot \nabla )v_{f}$ represents the advective acceleration term resulting from non-uniform velocity distribution, and this term is characterized by its non-linearity. On the right-hand side of the equation, divΓ signifies the divergence of stress tensor, which accounts for surface forces acting on a unit volume of fluid; F denotes mass force acting on a unit volume of fluid.

The fluid under consideration is water, which can be characterized as a homogeneous, viscous, and incompressible medium. In order to determine the stress tensor Γ, an appropriate expression is employed:(24)$$\Gamma =-pI+\mu_{f}(\nabla v_{f}+{\left(\nabla v_{f}\right)}^{T}),$$

The fluid pressure, represented by −pI in this equation, corresponds to the average of the three normal stress components p multiplied by the unit tensor I. The dynamic viscosity of the fluid is denoted as $\mu_{f}$, while $\mu_{f}(\nabla v_{f}+{\left(\nabla v_{f}\right)}^{T})$ represents the viscous stress tensor.

The solid control equation is utilized to describe the deformation of solids and solve stresses. In this project, the biomimetic fish is simplified as a homogeneous isotropic linear elastic material structure. The momentum conservation equation for the solid structure can be expressed as follows:(25)$$\rho_{s}{\ddot{d}}_{s}=divS+F_{v}\;\;\;\;\;\;\mathrm{in}\;\Omega_{X},$$

The equation describes the relationship between various parameters: $\rho_{s}$ represents the density of the solid material, ${\ddot{d}}_{s}$ denotes the vector representing the added velocity of the solid domain in the spatial coordinate system, $F_{v}$ signifies the mass force acting on a unit volume of solid, and S refers to the reference force tensor (Lagrangian stress tensor). The calculation formula for determining S is given as follows:(26)$$S=F^{e}S^{e}F^{0*},$$



(27)
$$S^{e}=2\mu_{s}E^{e}+\lambda tr(E^{e})I$$



The pre-stress field $F_{0}$ (with its adjoint matrix denoted as $F^{0*}$) is defined, while $F_{e}=FF_{0}^{-1}$ represents the elastic deformation field. $U_{s}$ denotes the solid displacement, and $F=I+\nabla U_{s}$. The elastic strain tensor $E^{e}=E^{1}-E^{0}$, where $E_{1}$ and $E_{0}$ are the nonlinear elastic strain tensors (Green–Lagrange strain tensors) E and $E_{0}$ are calculated as follows:(28)$$\left\{\begin{array}{l} E=\frac{1}{2}(F^{T}F-I)\\ E^{0}=\frac{1}{2}(F^{0T}F^{0}-I) \end{array}\right.,$$

$\mu_{s}$ and λ are the Lamé constants, and their formula is:(29)$$\left\{\begin{array}{l} \mu_{s}=\frac{Ev}{\left(1+v)(1-2v\right)}\\ \lambda =\frac{E}{2(1+v)} \end{array}\right.,$$

The equation defines E as the modulus of elasticity for the equivalent solid material, while v represents the Poisson’s ratio of said material.

The boundary conditions are solely determined by the problem at hand and remain unaffected by different reference configurations. The fish body’s curved motion is simulated through a time-varying nonlinear strain tensor $E^{0}=E^{0}(x,y,t)$. When $sym\nabla U_{s}=F^{0}(sym\nabla U_{s}=\nabla {U_{s}}^{T}+\nabla U_{s})$, the applied nonlinear strain tensor $E^{0}$ remains compatible, potentially resulting in no pressure generation during the horizontal rocking of the fish body. However, the counteracting force exerted by the surrounding flow field acts upon the fish body’s horizontal rocking at any given moment, necessitating the inclusion of dynamic boundary conditions:(30)Tn=−Γn,

The normal vector n represents the fluid–solid coupled boundary, while $T=S{\left(F^{*}\right)}^{-1}$ denotes the true stress (Cauchy stress) tensor of the solid. Additionally, Γ refers to the surface stress tensor of the fluid.

The kinematic boundary condition stipulates that the rate of deformation of the solid at the interface between fluid and structure is equivalent to the velocity of the fluid.
(31)$$v_{f}={\dot{u}}_{s},$$

The solid walls are set as boundaries around the flow field, and during the autonomous swimming of the biomimetic fish, the far field remains undisturbed with no inflow. Consequently, a no-slip condition is imposed on the wall boundary to ensure accurate simulation:(32)$$v_{f}=0,$$

The computational domain grid in the FSI interface environment is continuously adjusted to conform to the current shape of the fluid domain through the implementation of the ALE method.

### 4.2. Dynamic Simulation Results and Discussion

The biomimetic fish, as depicted in Figure 15, exhibits a rightward swimming motion with a flapping frequency of 1 Hz and an amplitude of 100 mm. Contraction is represented by the red section of the fish body, while extension is denoted by the blue section. The corresponding strain values are provided in the figure’s legend, with a maximum value of 0.17. An analysis of Figure 15 reveals that the blunt circular shape at the fish head position does not induce a reverse Kármán vortex street formation. In case such a phenomenon were to occur, it would generate fluid jets opposing the fish’s forward movement and consequently lead to drag generation. Hence, adopting a blunt circular shape for the fish head proves advantageous in minimizing drag during the undulatory swimming mode.

The simulation results demonstrate that, when the fish body undergoes symmetrical lateral oscillation based on the equation of the fish body wave motion, the bionic fish’s forward trajectory deviates from a straight line and instead follows an approximately S shaped curve towards the positive x axis. In the spatial coordinate system, the line segment connecting the center of mass of the fish body to the origin represents its swimming direction, while angle α between this line segment and the positive x axis signifies its swimming orientation. Figure 16 illustrates how α varies with time at a swinging frequency of 1 Hz. During the initial acceleration phase, movement is primarily driven by the y component of combined velocity in which significant directional changes occur. As the fish gradually enters into a cruising state, movement is predominantly governed by its x component of combined velocity; consequently, angle α exhibits certain amplitude and frequency fluctuations representing variations in swimming direction. Due to the initial deviation from positive x axis during acceleration phase, angle α remains positive even in the cruising state.

The simulation results demonstrate that the blunt-curved shape of the bionic fish’s head effectively mitigates vortex separation in the forward direction during oscillation, thereby facilitating drag reduction.

## 5. Experimental Testing

By comparing the deformation curve of the flexible spine in the bionic fish prototype with the simulated trajectory of the fish body’s centerline, we can verify the accuracy of the kinematic model for the fish body. Similarly, by comparing the position and posture of the tail fin in the bionic fish prototype with its trajectory at different time points, we can validate the correctness of our kinematic model for the tail fin.

The rigid head part of the bionic fish prototype was fixed to drive the flexible fish body and tail fin to swing in a controlled manner. A camera positioned directly above the bionic fish recorded its motion posture, while video editing software extracted feature time images for analysis. By comparing the mesh size (50 mm spacing) in these images, we were able to determine the swing of each point on the flexible spine and calculate the rotation angle of the tail fin around its hinges. Since the fish body’s motion is periodic and symmetrically swinging in linear mode, only select characteristic time positions and postures within half a cycle starting from 0 are presented, as shown in Figure 17.

In Figure 17, the fish body swing frequency is 1 Hz, and the maximum value of the pull-line variable $q_{\max}$ is 10 mm. Figure 17a–e demonstrate that at each characteristic moment, the swing amplitude of every point on the flexible spine of the prototype matches that of its corresponding point on the simulation curve. Additionally, both the position and posture of the tail fin align with those obtained from simulation results. Furthermore, it can be observed that the motion phase of the tail fin consistently leads by 90° compared to that of the fish body. These findings validate both our kinematics models for the fish body and the tail fin established in this study.

The bionic fish prototype is placed in a glass tank for swimming tests. The frequency of the fish’s body oscillation is set at 1 Hz, with an amplitude of 100 mm. In accordance with the linear motion mode, Figure 18 illustrates the sequential images capturing the swimming behavior of the bionic fish prototype.

The swimming capabilities of the bionic fish were primarily assessed in this section. The test results indicate that the overall weight of the bionic fish meets the design requirements, and its body density is slightly lower than that of water. The immersion depth of the fish can be adjusted by adding weight blocks, ensuring an excellent underwater sealing performance with no leakage during swimming. Moreover, the bionic fish prototype demonstrates an ability to swim forward in a designed motion posture while increasing both swing frequency and swimming speed.

## 6. Conclusions

This paper presents a novel cable-driven flexible spinal mechanism and completes the overall design scheme of a biomimetic fish. An experimental prototype was fabricated, and its kinematic model was established using the segmental constant curvature method for the flexible spinal fish body. The linear relationship between cable variables and fish body oscillation was solved through MATLAB simulation analysis. Additionally, an analytical method was employed to obtain the linear relationship expression between the cable variables and the tail fin hinge angle. A comparison is made between the centerline of the flexible spinal biomimetic fish and the real fish body wave curve to assess their degree of fitting. Simulation results demonstrate that under similar oscillation frequency conditions, the joint-chain type biomimetic fish exhibits better fitting when the oscillation amplitude is small, while the degree of fitting for the flexible spinal biomimetic fish improves as oscillation amplitude increases gradually. Furthermore, a two-dimensional numerical model, based on the elastic mechanic’s principles in ALE description and fluid mechanics principles, is developed to simulate the autonomous swimming behavior of this biomimetic fish; fluid–structure coupling control equations are also established accordingly. The simulation results reveal that, due to its rounded concave shape, vortices formation in front of this fish during the oscillation process can be avoided effectively, leading to reduced drag force generation. These simulation findings provide a theoretical basis for control optimization purposes. Moreover, a comparison with the actual motion process of the prototype fish validates the correctness of the motion model as well as the feasibility of the design scheme; underwater experiments serve as principle test experiments which confirm good waterproof sealing capability and the achievement of the desired weight target along with capabilities for linear motion and turning. Based on the future trends in bionic fish development, the following prospects are proposed for the in-depth research and development of the next generation of bionic fish: The first-generation bionic fish had a large head, which made it difficult to achieve precise control over its center of gravity and buoyancy. Therefore, further optimization of its body shape is necessary. While the main focus of research for the first-generation bionic fish was on middle-layer motion control and swimming performance, attention must now shift towards studying upper-layer control based on the motion model presented in this paper. This includes task planning, trajectory planning, navigation fault-tolerant control, coordination control among groups of fish, and obstacle avoidance control. The kinematic method used to control movement in first-generation bionic fish lacked adaptability; therefore, future research should develop a CPG-based method that can make these devices more intelligent with adaptive functions.

## Figures and Tables

**Figure 1 biomimetics-09-00345-f001:**
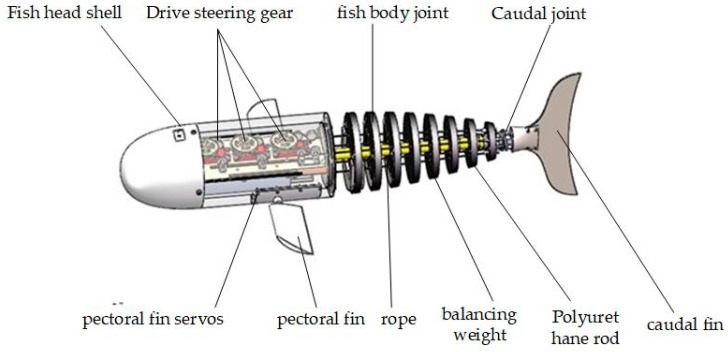
The Overall Structure of a Bionic Fish.

**Figure 2 biomimetics-09-00345-f002:**
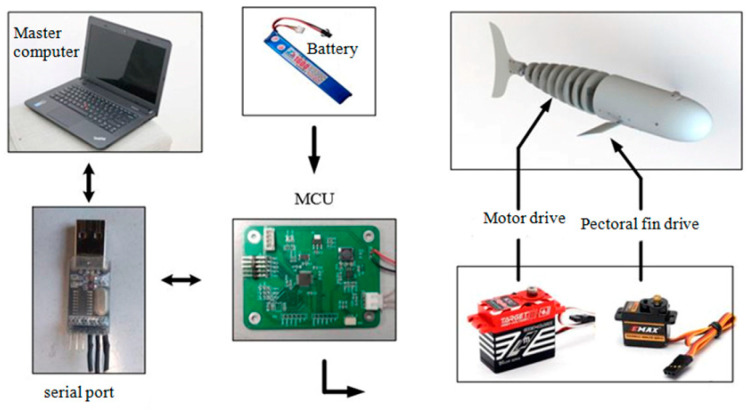
Hardware composition of control system.

**Figure 3 biomimetics-09-00345-f003:**
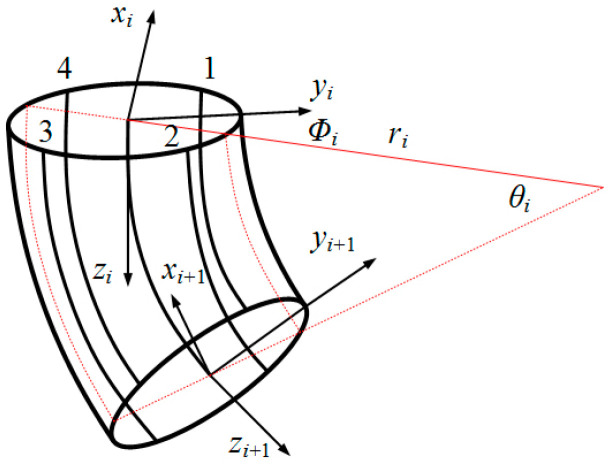
A piecewise curvature model of flexible fish body.

**Figure 4 biomimetics-09-00345-f004:**
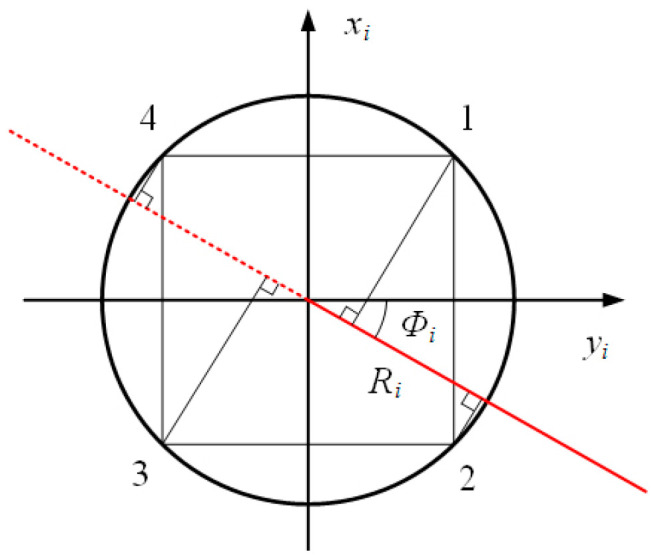
Geometric relationship in driving line of section i.

**Figure 5 biomimetics-09-00345-f005:**
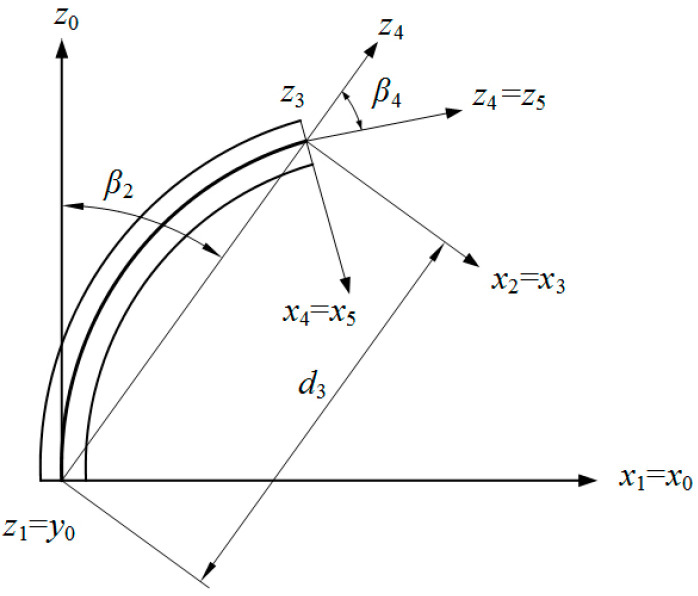
Adjacent coordinate transformation process.

**Figure 6 biomimetics-09-00345-f006:**
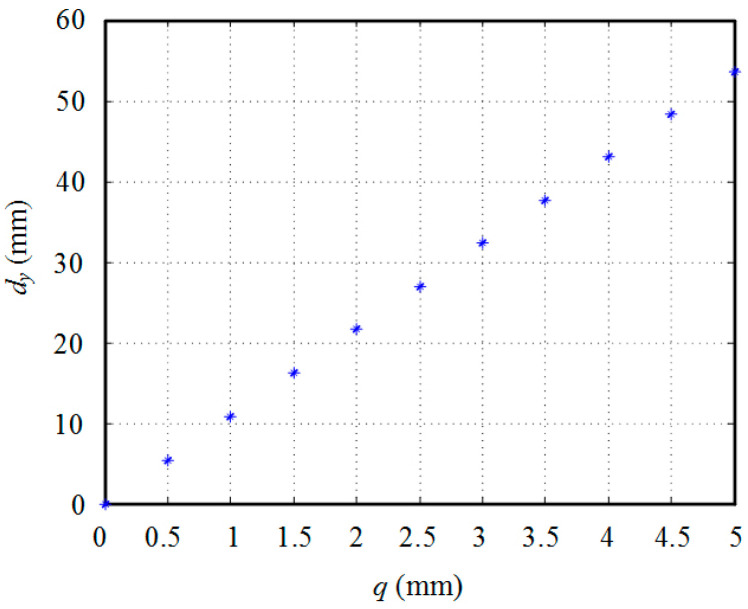
Driving Line Variables and Fish Body End Swing.

**Figure 7 biomimetics-09-00345-f007:**
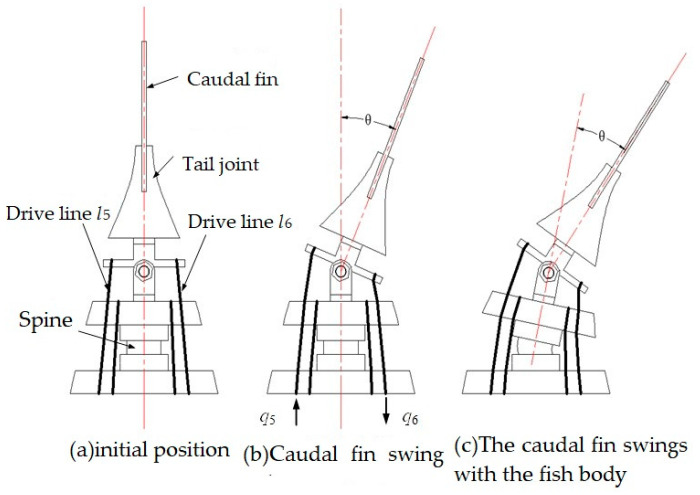
Top view of tail fin motion.

**Figure 8 biomimetics-09-00345-f008:**
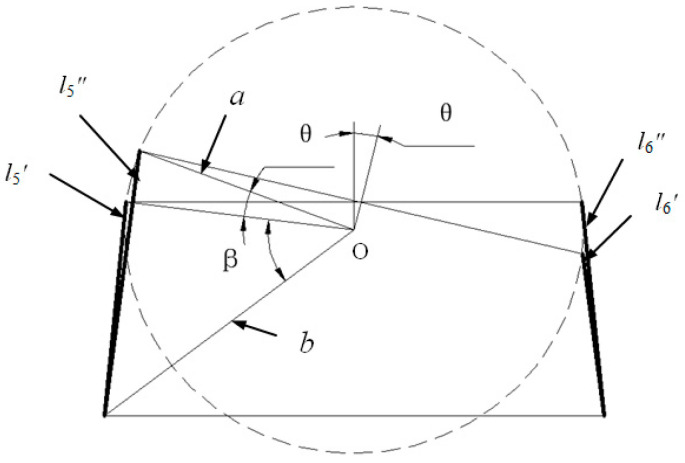
The relationship between the driving line variables and the tail fin swing angle.

**Figure 9 biomimetics-09-00345-f009:**
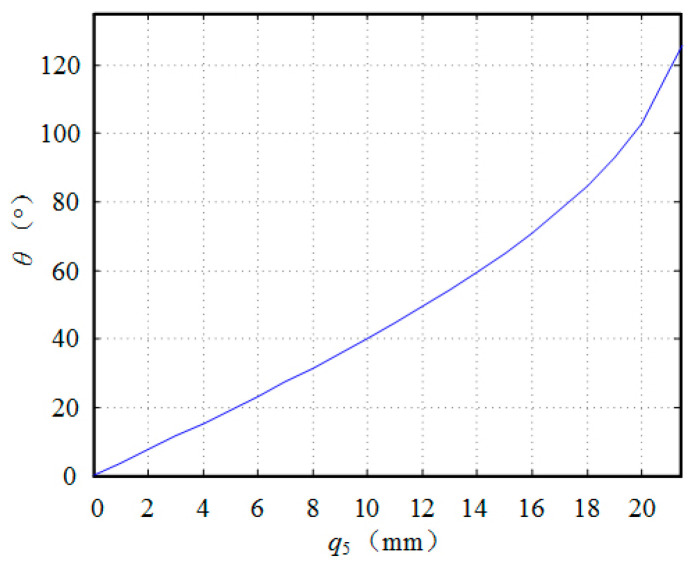
The relationship between the tail fin pull line and the angle of rotation.

**Figure 10 biomimetics-09-00345-f010:**
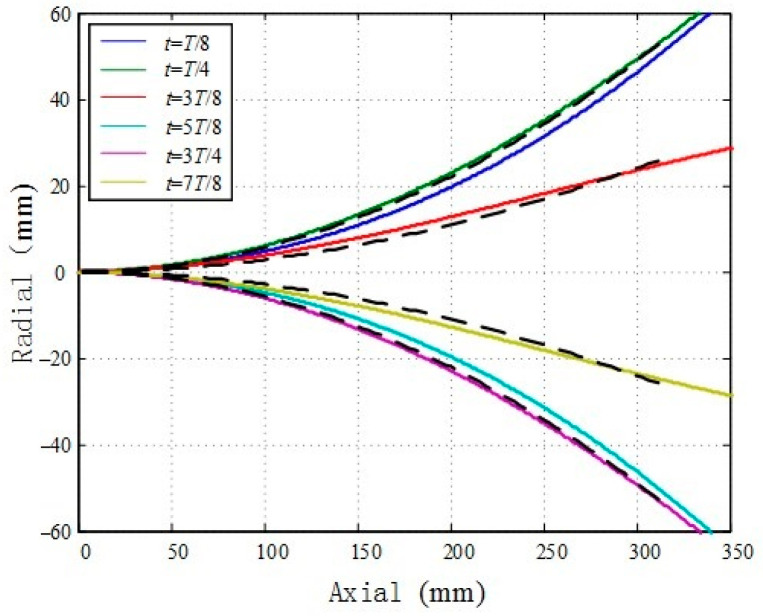
The centerline of fish body and the curve pair of fish body wave.

**Figure 11 biomimetics-09-00345-f011:**
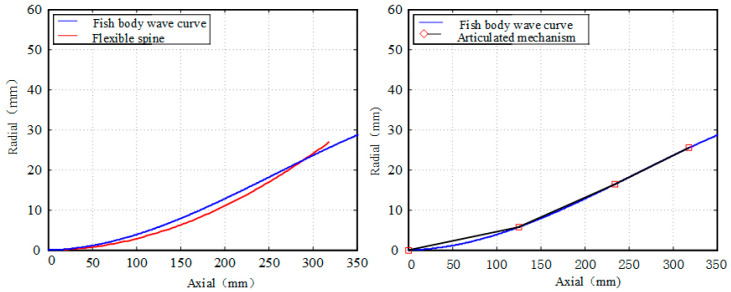
T/8 time fitting comparison.

**Figure 12 biomimetics-09-00345-f012:**
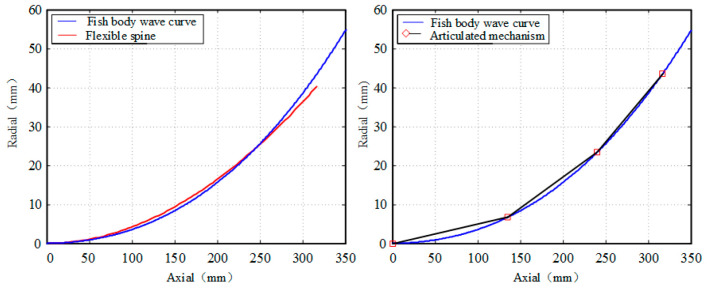
3T/16 time fitting comparison.

**Figure 13 biomimetics-09-00345-f013:**
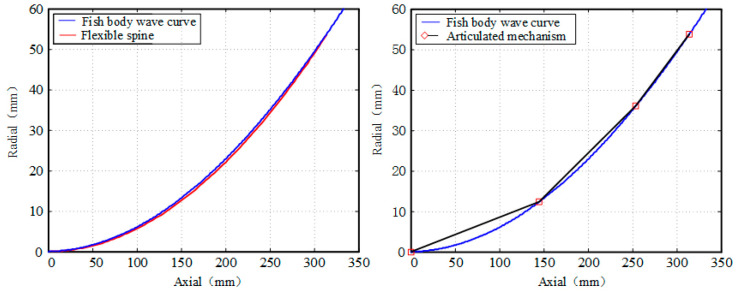
T/4 time fitting comparison.

**Figure 14 biomimetics-09-00345-f014:**
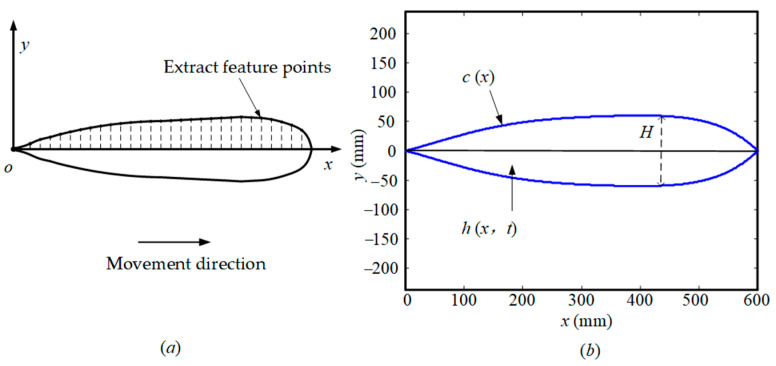
The contour of the fish body. (**a**) Bionic fish contours projected in the x-y plane; (**b**) Parametric Expression Curve for Bionic Fish Contours.

**Figure 15 biomimetics-09-00345-f015:**
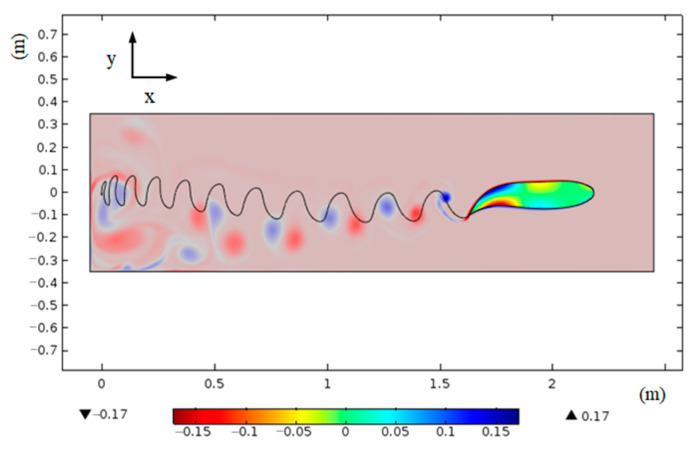
Wake and fish body deformation.

**Figure 16 biomimetics-09-00345-f016:**
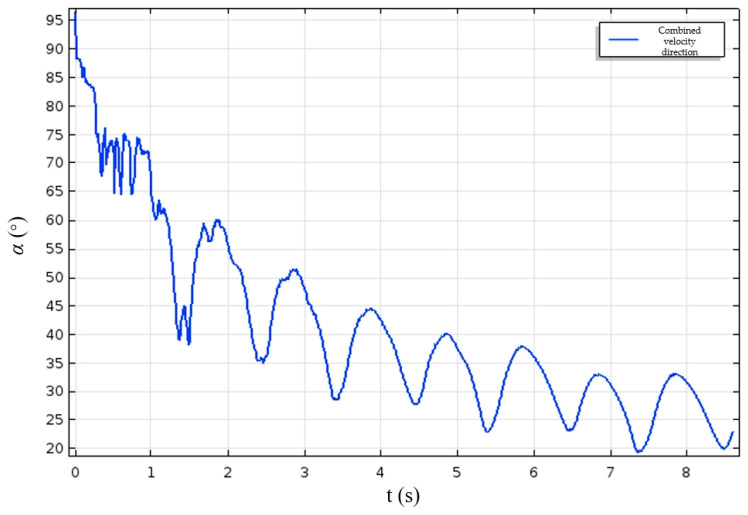
The direction of the fish’s body axis while swimming.

**Figure 17 biomimetics-09-00345-f017:**
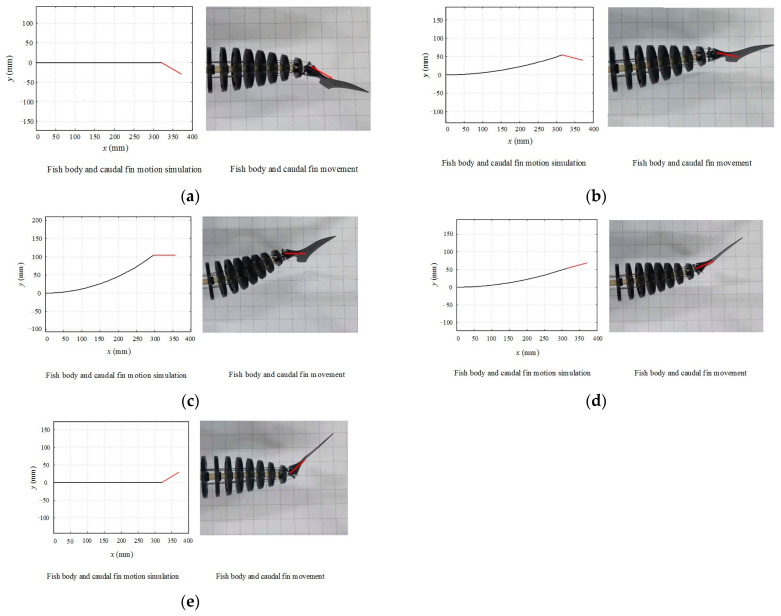
Comparisons of motion curves of fish. (**a**) *t* = 0*T*; (**b**) *t* = *T*/8; (**c**) *t* = *T*/4; (**d**) *t* = 3*T*/8; (**e**) *t* = *T*/2.

**Figure 18 biomimetics-09-00345-f018:**
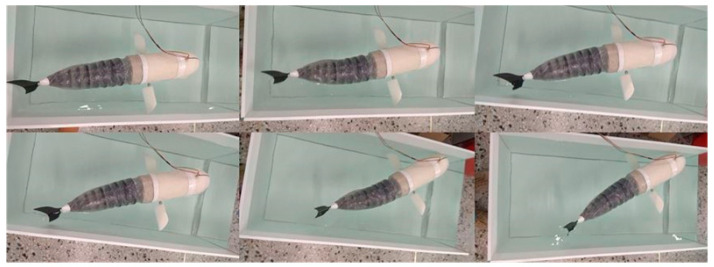
Bionic Fish Linear Motion Sequence Diagram.

**Table 1 biomimetics-09-00345-t001:** Bionic Fish Size Parameters.

Framework	Sizes (mm)
Fish head	320 × 120 × 133
Fish body	320 × 112 × 130
Tail steering gear	40.5 × 20.5 × 36
Tail fin length	63
Pectoral steering gear	23 × 11.5 × 24
Control module	164 × 70 × 20
Total length	703
Head-to-tail ratio	16:19

**Table 2 biomimetics-09-00345-t002:** D–H Transform Parameters.

Motion Order	β	d	a	α
1	$\Phi_{i}$	0	0	−π/2
2	$\theta_{i}/2$	0	0	−π/2
3	0	$2r_{i}\sin(\theta_{i}/2)$	0	−π/2
4	$\theta_{i}/2$	0	0	−π/2
5	$-\Phi_{i}$	0	0	−π/2

**Table 3 biomimetics-09-00345-t003:** Envelope area of centerline and body wave of biomimetic fish $\left(mm^{2}\right)$.

Envelope Area	*T*/8	3*T*/16	*T*/4
$S_{1}$	322.25	235.33	162.75
$S_{2}$	119.26	251.47	371.15

## Data Availability

The datasets used or analyzed during the current study are available from the corresponding author on reasonable request.

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
