# Peer review of "Kinematic Modeling and Experimental Study of a Rope-Driven Bionic Fish"

_biomimetics, 2024, doi:10.3390/biomimetics9060345_

Round 1
Reviewer 1 Report
Comments and Suggestions for Authors
The paper needs major revisions. There are many grammatical errors in the paper. Figures and equations in the paper are not good format.
Acknowledge any challenges or limitations associated with your proposed system. Kinematics of the bionic fish is not essential for underwater. Dynamic model of the bionic fish should be developed. This model should be verified by experimental results. Contributions of the paper should be stated clearly.
Discuss potential real-world applications and scenarios where your proposed system could be particularly beneficial. For instance, consider different underwater environments and situations (Wavy sea and current sea).
Comments on the Quality of English Language
There are many grammatical errors in the paper. Paper's english should be enhanced.
Author Response
Thank you very much for taking the time to review this manuscript. Please find the detailed responses below and the corresponding revisions/corrections highlighted/in track changes in the re-submitted files
Point-by-point response to Comments and Suggestions for Authors
Comments 1: The paper needs major revisions. There are many grammatical errors in the paper. Figures and equations in the paper are not good format.
Response 1: Thank you for pointing this out. We agree with this comment. Therefore, we have revised the entire paper and modified the table and figure formats according to the journal's formatting guidelines. We have also used the Math Type 6.9 tool to correct any issues with the formulas.
Comments 2: Acknowledge any challenges or limitations associated with your proposed system. Kinematics of the bionic fish is not essential for underwater. Dynamic model of the bionic fish should be developed. This model should be verified by experimental results. Contributions of the paper should be stated clearly.
Response 2: Thank you for pointing this out. We agree with this comment. Therefore.Therefore:
2.1 We have added new research content and its challenges and future prospects (from lines 562 to 573 on page 19 to lines 573 on page 20, in red).
2.2 We have moved the content of Chapter 3 from the original article to Chapter 3.1 of this article, and the content of Chapter 4 from the original article to Chapter 3.2 of this article. We have also added a new Chapter 4 on the dynamics analysis of bionic fish (13 pages from line 359 to line 501, in red).
2.3 We have added a new section on the contributions of the paper (pages 2, lines 64-72, in red) in the article.
Comments 3:Discuss potential real-world applications and scenarios where your proposed system could be particularly beneficial. For instance, consider different underwater environments and situations (Wavy sea and current sea).
Response 3: We have added a new section on the applications of the research presented in this paper (from lines 64 to 72 on page 2, in red).
Finally, we revised the entire article and paid special attention to polishing the English language.
Reviewer 2 Report
Comments and Suggestions for Authors
In this paper, the authors have presented a tendon-driven bionic robotic fish. The fish robot consists of a constant curvature-based flexible spine, motor, and control module. The study includes the development of a prototype, the establishment of a kinematic model, and validation through simulation and experimental results. The experimental tests confirm the effectiveness of the design, highlighting its potential for applications in underwater exploration. However, the paper is poorly written, the images are not clear, and it is difficult to understand what the novelty of the work is. The flexible spine-based tendon-driven work has already been reported previously.
Here are my major comments:
1. What is the novelty of this work? This work focuses on a flexible spine and tendon-driven system. Similar work has been reported by Qiu, C., Wu, Z., Wang, J., Tan, M., and Yu, J., 2022. "Locomotion optimization of a tendon-driven robotic fish with variable passive tail fin." IEEE Transactions on Industrial Electronics, 70(5), pp.4983-4992.
2. The introduction is poorly written and insufficient. There is a lot of work around tendon-driven fish robots. Please rewrite the introduction and highlight key contributions of previous works.
3. It is unusual to see the authors have added the university name while citing papers. Usually, it is just the author's name, for example, "name et al. (year)." Please check the journal’s formatting rules.
4. The reviewer suggests moving Figure 3 to the side of Figure 1, making it easier to reference.
5. There is a spelling mistake in line 88, “micor controller,” which should be “microcontroller.”
6. Please check the following papers that used a piecewise kinematic model:
- Mishra, A.K., Mondini, A., Del Dottore, E., Sadeghi, A., Tramacere, F., and Mazzolai, B., 2018. "Modular continuum manipulator: analysis and characterization of its basic module." Biomimetics, 3(1), p.3.
- Webster III, R.J., and Jones, B.A., 2010. "Design and kinematic modeling of constant curvature continuum robots: A review." The International Journal of Robotics Research, 29(13), pp.1661-1683.
7. There are many places with symbol subscript issues. Please correct them throughout the paper.
8. It is difficult to imagine the robotic fin movement. Please provide a clearer illustration.
9. Please include a comparative table of different fish robots to demonstrate the novelty of your work.
10. Please include a video of the robot in action.
Comments on the Quality of English LanguagePlease edit the paper thoroughly as there are grammatical and punctuation errors in several places. It is difficult to read.
Author Response
Thank you very much for taking the time to review this manuscript. Please find the detailed responses below and the corresponding revisions/corrections highlighted/in track changes in the re-submitted files
Point-by-point response to Comments and Suggestions for Authors
Comments 1:What is the novelty of this work? This work focuses on a flexible spine and tendon-driven system. Similar work has been reported by Qiu, C., Wu, Z., Wang, J., Tan, M., and Yu, J., 2022. "Locomotion optimization of a tendon-driven robotic fish with variable passive tail fin." IEEE Transactions on Industrial Electronics, 70(5), pp.4983-4992.
Response 1: Thank you for pointing this out. We agree with this comment. Our study of biomimetic fish draws on the research presented in the paper you cited. However, our research introduces a novel approach by utilizing a flexible parallel mechanism with a fish-shaped structure to drive the biomimetic fish's flexible body. The internal spinal column of the fish body is composed of a single polyurethane rod with a low modulus of elasticity, enabling large bending deformations when driven by pulling ropes and accurately simulating the body deformation observed in Scombridae family of fish. The control system for the fish body resembles that of a pulling-rope-driven flexible body manipulator. Compared to serial-joint type mechanisms, our flexible parallel mechanism with pulling-rope drive provides greater flexibility in movement, equivalent to having an infinite number of serial joints. Additionally, we have incorporated an additional tail joint into our biomimetic fish design, adding one degree of freedom and enhancing swimming flexibility.
Comments 2: The introduction is poorly written and insufficient. There is a lot of work around tendon-driven fish robots. Please rewrite the introduction and highlight key contributions of previous works.
Response 2: Thank you for pointing this out. We agree with this comment. Therefore, we have revised the introduction, please refer to the latest draft for the Introduction.
Comments 3: It is unusual to see the authors have added the university name while citing papers. Usually, it is just the author's name, for example, "name et al. (year)." Please check the journal’s formatting rules.
Response 3: Thank you for pointing this out. We agree with this comment. Therefore, we also modified the citation format of the articles according to the journal's citation style guidelines, removing unnecessary university names.
Comments 4: The reviewer suggests moving Figure 3 to the side of Figure 1, making it easier to reference.
Response 4: Thank you for pointing this out. We agree with this comment. Therefore, we integrated Figure 3 and Figure 1 based on the content of the article, ultimately combining them into Figure 1, and then deleted Figure 3. Please refer to page 3, line 109 of the article.
Comments 5:There is a spelling mistake in line 88, “micor controller,” which should be “microcontroller.”
Response 5: Thank you for pointing this out. We agree with this comment. Therefore, we corrected the spelling of the word "microcontroller".
Comments 6: Please check the following papers that used a piecewise kinematic model:
- Mishra, A.K., Mondini, A., Del Dottore, E., Sadeghi, A., Tramacere, F., and Mazzolai, B., 2018. "Modular continuum manipulator: analysis and characterization of its basic module." Biomimetics, 3(1), p.3.
- Webster III, R.J., and Jones, B.A., 2010. "Design and kinematic modeling of constant curvature continuum robots: A review." The International Journal of Robotics Research, 29(13), pp.1661-1683.
Response 6: Thank you for pointing this out. We agree with this comment. Therefore, we have carefully reviewed the two articles you submitted and cited them in the Introduction. Please refer to lines 57 and 60 on page 2 of the article.
Comments 7:There are many places with symbol subscript issues. Please correct them throughout the paper.
Response 7: Thank you for pointing this out. We agree with this comment. Therefore,we checked the entire article and used the MathType 6.9 tool to correct the subscript symbol issues.
Comments 8: It is difficult to imagine the robotic fin movement. Please provide a clearer illustration.
Response 8:Thank you for pointing this out. We agree with this comment. Therefore,Based on the new Figure 1, we use language to describe the movement of the machine fish fins more clearly. The description is found on page 3, lines 98-108 of the article.
Comments 9:Please include a comparative table of different fish robots to demonstrate the novelty of your work.
Response 9:Thank you for pointing this out. We agree with this comment. Therefore,we have added a table for comparisons of different fish-like robots and the novelty of the research content, which is placed in Appendix A.The content is on pages 20 to 21, from line 586 to line 594.
Comments 10: Please include a video of the robot in action.
Response 10:Thank you for pointing this out. We agree with this comment. Therefore,we have sent the video of the machine fish to Assistant Editor - Doreen Li via email attachment.
Finally, we revised the entire article and paid special attention to polishing the English language.
Round 2
Reviewer 2 Report
Comments and Suggestions for Authors
The paper has improved after the revision.
Comments on the Quality of English LanguageIt looks readable, but it's definitely worth checking with another eye for any issues.